# Open-world Instance Segmentation: Top-down Learning with Bottom-up Supervision

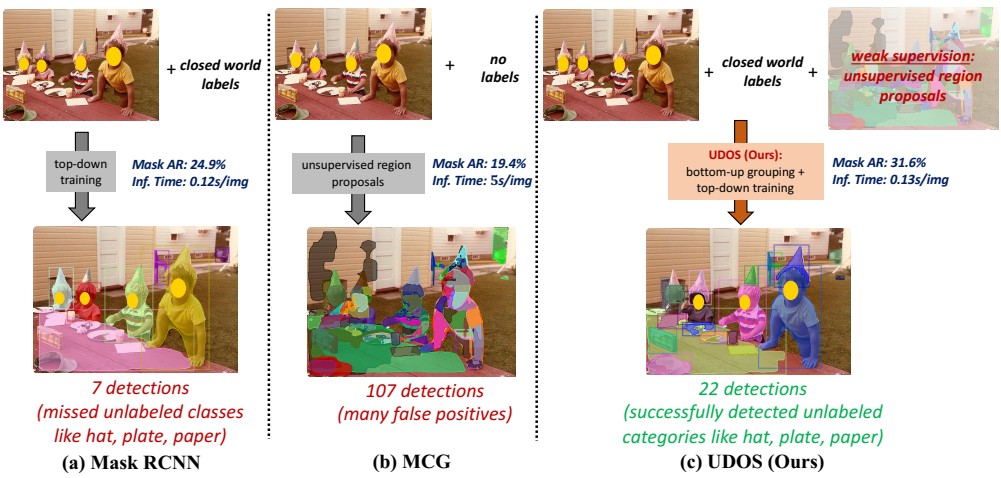

Fig. 1: **Open world segmentation using UDOS**. Image from COCO. **(a)** Mask R-CNN (He et al., 2017), trained on VOC-categories from COCO, fails to detect many unseen categories due to seen-class bias; **(b)** MCG (Pont-Tuset et al., 2016) provides diverse proposals, but predicts many over-segmented false-positives with noisy boundaries; **(c)** combining the advantages of (a) and (b) into a joint framework, UDOS efficiently detects unseen classes in open world when trained only using VOC-categories from COCO, while adding negligible inference time overhead.

## Abstract

Top-down instance segmentation architectures excel with predefined closed-world taxonomies but exhibit biases and performance degradation in open-world scenarios. In this work, we introduce bottom-**U**p and top-**D**own **O**pen-world **S**egmentation (UDOS), a novel approach that combines classical bottom-up segmentation methods within a top-down learning framework. UDOS leverages a top-down network trained with weak supervision derived from class-agnostic bottom-up segmentation to predict object parts. These part-masks undergo affinity-based grouping and refinement to generate precise instance-level segmentations. UDOS balances the efficiency of top-down architectures with the capacity to handle unseen categories through bottom-up supervision. We validate UDOS on challenging datasets (MS-COCO, LVIS, ADE20k, UVO, and OpenImages), achieving superior performance over state-of-the-art methods in cross-category and cross-dataset transfer tasks. Our code and models will be publicly available.

## 1 Introduction

Open world instance segmentation (Wang et al., 2021b) is the task of predicting class-agnostic instance masks for all objects within an image. A pivotal challenge therein lies in effectively segmenting novel instances, i.e., instances from categories not in the training taxonomy. This capability assumes paramount importance for ensuring the robust and dependable real-world deployment of instance segmentation models across domains like robotics (Xie et al., 2021), autonomous driving (Neuhold et al., 2017; Cordts et al., 2016), and embodied AI (Savva et al., 2019), where novel

objects are encountered regularly. While expanding taxonomy during annotation is a potential countermeasure, it presents notable challenges: it necessitates substantial human effort to amass sufficient annotations for each category, and achieving a comprehensive taxonomy encompassing *all* conceivable categories remains an impractical endeavor. Consequently, the emphasis remains on the model's capacity to generalize and proficiently segment novel objects—a more pragmatic approach.

Common instance segmentation frameworks like Mask R-CNN (He et al., 2017) often tightly couple recognition and segmentation (Wang et al., 2021b), making it challenging to accurately segment objects not present in the training data. This issue is particularly pronounced when these frameworks are trained with non-exhaustive annotations like MS-COCO (Lin et al., 2014), where out-of-taxonomy objects are treated as background, resulting in penalties for predictions made on these objects. In Fig. 1(a), a typical Mask R-CNN model, trained on the 20 VOC classes from the COCO dataset, effectively identifies objects within the training taxonomy such as people and chairs. However, it struggles to detect objects beyond this taxonomy, like hats, paper, and plates.

Conversely, classical bottom-up segmentation methods (Uijlings et al., 2013; Pont-Tuset et al., 2016; Grundmann et al., 2010) are class-agnostic and unsupervised by design, making them suitable for open-world scenarios. These methods rely solely on low-level cues such as shape, size, color and texture to generate object masks. However, they often suffer from over-segmentation, lacking a semantic understanding of objectness. In Fig. 1(b), MCG (Pont-Tuset et al., 2016) generates over-segmentation of objects with noisy boundaries.

How can we combine the strengths of both paradigms? We answer this question with our novel approach for open-world instance segmentation, termed UDOS (Bottom-Up and Top-Down Open-World Segmentation). UDOS seamlessly integrates the advantages of both top-down and bottom-up methods into a unified and jointly trainable framework. UDOS (Fig. 1c) effectively segments known categories like persons and chairs while demonstrating robust generalization to unseen categories like party hats, paper, glasses, and plates.

UDOS is grounded on two key intuitions: First, we recognize the value of weak supervision from class-agnostic segmentation generated by unsupervised bottom-up methods (Uijlings et al., 2013; Pont-Tuset et al., 2016; Grundmann et al., 2010). This supplementation complements potentially incomplete human annotations, ensuring holistic image segmentation without designating any region as negative. Second, we leverage seen-class supervision to bootstrap objectness, introducing an affinity-based grouping module to merge parts into complete objects and a refinement module to enhance boundary quality. Despite only being trained on seen categories, we observe that both part-level grouping and boundary refinement generalize well to novel categories.

UDOS is the first approach that effectively combines top-down architecture and bottom-up supervision into a unified framework for open-world instance segmentation, and we show its superiority over existing methods through extensive empirical experiments. Our contributions are:

1. We propose UDOS for open-world instance segmentation that effectively combines bottom-up unsupervised grouping with top-down learning in a single jointly trainable framework (Sec. 3).
2. We propose an affinity based grouping strategy (Sec. 3.2) followed by a refinement module (Sec. 3.3) to convert noisy part-segmentations into coherent object segmentations. We show that such grouping generalizes well to unseen objects.
3. UDOS achieves significant improvements over competitive baselines as well as recent open-world instance segmentation methods OLN (Kim et al., 2022), LDET (Saito et al., 2022) and GGN (Wang et al., 2022) on cross-category generalization (VOC to NonVOC) as well as cross-dataset (COCO to UVO, ADE20K and OpenImagesV6) settings (Sec. 4).

## 2 RELATED WORKS

**Object detection and instance segmentation.** In the past, these tasks relied on handcrafted low-level cues with a bottom-up approach: graph-based grouping (Felzenszwalb & Huttenlocher, 2004; Grundmann et al., 2010), graph-based methods (Shi & Malik, 2000; Boykov et al., 2001; Felzenszwalb et al., 2008), deformable parts (Felzenszwalb et al., 2009), hierarchical and combinatorial grouping (Arbelaez, 2006; Pont-Tuset et al., 2016) or Selective Search (Uijlings et al., 2013). The rise of deep learning brought about top-down approaches, excelling in various detection and segmentation tasks, including object proposals (Pinheiro et al., 2015; Kuo et al., 2015), object detec-

Fig. 2: UDOS overview

Fig. 3: Proposed UDOS pipeline

**(Fig. 2) UDOS overview:** Training and inference phases in UDOS. The unsupervised proposal generation is only present during training and not used in inference. **(Fig. 3) Proposed UDOS pipeline:** During training, we first augment the ground truth annotations on seen classes ($S$) with masks provided by the unsupervised segmentation algorithm ($U$) and use it to supervise the part mask prediction head in (a) (Sec. 3.1). As these predictions might only correspond to part-segments on unknown classes (*head* of the horse, *body* of the dog), we use an affinity based grouping strategy in (b) that merges part segments corresponding to the same instance (Sec. 3.2 and Fig. 4). We then apply a refinement head in (c) to predict high-quality segmentation for complete instances.

tion (Ren et al., 2015), semantic segmentation (Long et al., 2015), instance segmentation (He et al., 2017; Arnab & Torr, 2017; Bolya et al., 2019) and panoptic segmentation (Kirillov et al., 2019; Wang et al., 2021a). However, this paper addresses a fundamentally different challenge. Instead of assuming a closed-world scenario where training and testing share the same taxonomy, we focus on the open-world, which involves segmenting both in-taxonomy and out-of-taxonomy instances. As in Fig. 1 and Sec. 4.2, top-down methods exhibit a bias towards seen classes and struggle to detect novel objects.

**Open world instance segmentation.** Open-world vision involves generalizing to unseen objects (Bendale & Boult, 2015; Pinheiro et al., 2015) and is regaining traction in computer vision (Hsu et al., 2020; Liu et al., 2019; Wang et al., 2021b; Liu et al., 2021; Joseph et al., 2021; Kim et al., 2022). We focus on open-world instance segmentation (Wang et al., 2021b), where the goal is to detect and segment objects, even if their categories weren't in the training data. This differs from works (Hu et al., 2018; Kuo et al., 2019) that rely on categories with bounding box annotations during training. Prior methods (Du et al., 2021; Pham et al., 2018; Dave et al., 2019; Jain et al., 2017) often used additional cues like video, depth, or optical flow. In contrast, UDOS requires no extra annotations and relies on unsupervised proposal generation. Our work is related to (Wang et al., 2021b; Kim et al., 2022; Wang et al., 2022). Wang et al. (2021b) introduces a benchmark, while our paper presents a novel approach. OLN (Kim et al., 2022) enhances objectness but uses only seen-class annotations in training. UDOS combines top-down training and bottom-up grouping for novel object segmentation. GGN (Wang et al., 2022) shares a similar approach with UDOS by using bottom-up grouping. However, GGN uses pixel-level pairwise affinities for grouping, while UDOS uses part-level pairwise affinities, with fundamentally different grouping principles. Our results show that UDOS performs well compared to GGN, indicating it could be a complementary method.

**Combining bottom-up and top-down.** Recent research has revisited bottom-up methods in representation learning (Hénaff et al., 2021; Bar et al., 2021; Zhang & Maire, 2020). In instance segmentation, bottom-up grouping has improved local segmentation quality using affinity maps (Liu et al., 2018; Bailoni et al., 2019; Xu et al., 2020; Wang et al., 2022) or pixel grouping (Liu et al., 2017; Hwang et al., 2019; Siam et al., 2021). However, these approaches focus on closed-world taxonomies. Our work combines top-down training and bottom-up grouping for open-world instance segmentation, distinguishing it from prior grouping-based methods (Kong & Fowlkes, 2018; Ahn & Kwak, 2018) that use low-level pixel features. We also address open-world instance segmentation, unlike prior work on 3D part discovery or shape analysis (Luo et al., 2020).

## 3 PROPOSED METHOD

**Problem definition**. Given an image $I \in \mathbb{R}^{H \times W \times 3}$, the goal of open world instance segmentation is to segment all object instances in $I$ regardless of their semantic categories, which includes objects that were both seen and unseen during training. Following prior works (Pinheiro et al., 2015; Kim

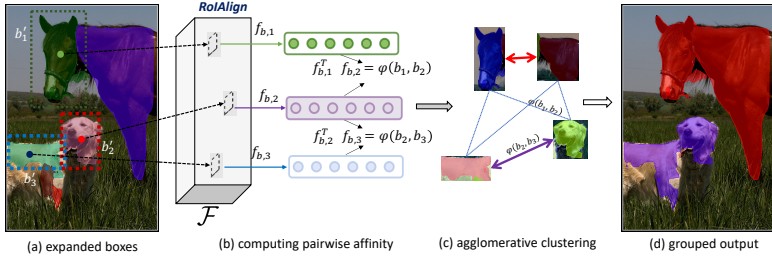

(a) expanded boxes     (b) computing pairwise affinity     (c) agglomerative clustering     (d) grouped output

Fig. 4: **Grouping module**. (a) the bounding boxes $b_i$ of the predicted part-masks are expanded to incorporate local context. (b) The features $f_{b,i}$ are extracted using RoIAlign operator on the FPN features $\mathcal{F}$ with the expanded bounding boxes $b_i'$, and are used to compute pairwise affinity $\phi(b_i, b_j)$ using cosine similarity. (c) A clustering algorithm is used to group parts into whole object instances, as shown in (d). Note that the inaccuracies in the output from grouping module are later corrected by the refinement module.

et al., 2022; Wang et al., 2022), we adopt class-agnostic learning strategy, in which all annotated classes are mapped to a single foreground class during training and predictions are class-agnostic.

**Method overview of UDOS**. We visualize the training and inference flows of UDOS in Fig. 2. UDOS consists of part-mask prediction (Fig. 3a), affinity-based grouping (Fig. 3b) and refinement (Fig. 3c). We use class-agnostic Mask R-CNN (He et al., 2017) with FPN (Lin et al., 2017) as backbone, and we denote the FPN feature map as $\mathcal{F}$.

### 3.1 PART-MASK PREDICTION

**Generating candidate object regions**. We start by creating weak supervision using unsupervised segmentation algorithms (e.g., selective search (Uijlings et al., 2013) or MCG (Pont-Tuset et al., 2016)) for each image in the training set. These segmentation masks are class-agnostic and cover the entire image, regardless of in-taxonomy or out-of-taxonomy objects. We intentionally favor over-segmentation during proposal generation by tuning the algorithms' hyperparameters (e.g., scale and $\sigma$ in selective search). It's important to note that this process is a *one-time* effort before training and is not needed during inference (Fig. 3).

**Augmenting labels using part-masks**. Next, for each training image $I$, we create a triplet $(I, S, U)$, where $S = \{s_i\}_{i=1}^{N_s}$ represents the set of ground truth box and mask labels for annotated categories, and $U = \{u_i\}_{i=1}^{N_u}$ represents masks generated by the unsupervised segmentation algorithm, offering more extensive but potentially noisy region proposals. We use the augmented masks set $A = S \cup U$ as supervision to train a top-down instance segmentation system, referred to as the part-mask prediction network. This network may predict only parts of objects in alignment with the provided supervision (output Fig. 3a). To avoid label duplication, we exclude part masks from $U$ that overlap with any ground truth mask in $S$ with an IoU greater than 0.9. In essence, while masks in $S$ provide information for detecting in-taxonomy classes, masks in $U$ assist in segmenting part masks for **all** objects, offering complementary training signals to the network. This strategy offers two key advantages over top-down training with only ground truth masks in $S$. First, unsupervised region proposals from $U$ account for un-annotated image regions that may contain valid out-of-taxonomy objects, preventing the network from mistakenly labeling them as background. Second, despite masks in $U$ potentially not representing complete objects, they still provide valuable training signals for detecting out-of-taxonomy objects. For example, accurately detecting parts of a dog, such as the head, body, and ears in Fig. 3, proves useful in the final segmentation of the entire dog through our part-mask grouping strategy.

### 3.2 GROUPING MODULE

To bridge the gap between mid-level part-masks (Sec. 3.1) and complete object instances, we propose an efficient lightweight grouping strategy to merge parts into objects. We compute pairwise affinities between features of the *expanded* parts, and cluster them based on affinities.

**Pairwise affinity** We denote the predictions made by the network in the first phase by $P = \{p_i\}_{i=1}^{n_p}$, where $n_p$ is the number of predictions, and $p_i$ contains mask ($m_i$) and box ($b_i$) predictions made on seen as well as unseen categories. For each bounding box $b_i \in p_i$, we first expand the width and height of the box by a factor $\delta(0 < \delta < 1)$ to compute a new, larger bounding box $b_i'$ (Fig. 4a).

$$b_i : (x_i, y_i, h_i, w_i) \xrightarrow{\text{expand}} b_i' : (x_i, y_i, (1 + \delta) * h_i, (1 + \delta) * w_i) \qquad (1)$$

where $(x_i, y_i)$ is the center and $(h_i, w_i)$ are the original height and width of box $b_i$. This inflation allows us to ingest useful context information around the part and the underlying object, better informing the affinities between the part-masks. Next, we compute the ROIAlign features for all the boxes $\{b_i'\}$ from the FPN feature map $\mathcal{F}$ resulting in a $d$-dim feature for each part-prediction denoted using $\{f_{b,i}\}_{i=1}^{n_p} \in \mathbb{R}^d$. The pairwise affinity between two part-predictions $(p_i, p_j) \in P$ is then computed using the cosine similarity between the corresponding feature maps (Fig. 4b).

$$\phi(p_i, p_j) = \frac{f_{b,i}^T \cdot f_{b,j}}{\|f_{b,i}\|\|f_{b,j}\|}; f_{b,i} = \text{RoIAlign}(\mathcal{F}, b_i') \qquad (2)$$

We visualize the parts retrieved using pairwise affinity for few examples in Fig. 6. While Wang et al. (2022) has shown strong generalization of pixel pairwise affinities, our novelty lies in showing that the part-mask pairwise affinities generalize better across object categories.

**Affinity based grouping** We use a clustering algorithm to merge parts based on the soft affinity scores given in equation 2. Our clustering objective can be formulated as follows:

$$\max_G \sum_{k=1}^{|G|} \sum_{p_i, p_j \in g_k} \phi(p_i, p_j), \quad \text{s.t.} \sum_{k=1}^{|G|} |g_k| = n_p \qquad (3)$$

where $G$ is a possible partition of the $n_p$ predictions, $|G|$ denotes the total number of partitions and $k^{\text{th}}$ partition in $G$ is denoted by $g_k$ $(1 \le k \le |G|)$. In other words, given a set of elements along with their pairwise affinity scores, our clustering algorithm produces a partition of the elements that maximizes the average affinities within each partition. We use an off-the-shelf agglomerative clustering algorithm from Bansal et al. (2004) provided by `scikit-learn` (Pedregosa et al., 2011). It is parameter-free, lightweight, and fast, incurring minimum time and memory overhead while clustering hundreds of part-masks in each iteration. As shown in Sec. 4.4 our final framework adds negligible inference time overhead to the backbone. We merge all the part masks (and boxes) within each partition group $g_k$ to form more complete masks, representing whole objects (Fig. 3b). Since the original predictions in $P$ might also represent whole objects on seen classes, we combine the originally detected masks as well as the grouped masks into our output at this stage.

### 3.3 REFINEMENT MODULE

To address potential blurriness in the grouped masks due to noisy initial segmentation, we incorporate a refinement module. Designed similar to the RoIHeads in Mask R-CNN, this module takes the predictions generated after the grouping stage as inputs (Fig. 3c). We train the refinement head exclusively on annotated ground truth instances from $S$ to introduce the concept of object boundaries into the predictions (only available in the annotated masks). We found that this boundary refinement also generalizes well to unseen categories. We jointly train the backbone and refinement heads in a single stage, using losses from the part-mask prediction and refinement modules.

**Objectness ranking** Following (Kim et al., 2022), we add box and mask IoU branches to our RoI-Head in part-mask predictions as well as refinement heads to compute the localization quality. IoU metrics are shown to improve objectness prediction (Huang et al., 2019) and avoid over-fitting to seen instances (Kim et al., 2022) when trained with non-exhaustive annotations. We use box and mask IoU heads with two fc-layers of 256-dim each followed by a linear layer to predict the IoU score, trained using an L1 loss for IoU regression.

**Inference** (Fig. 2). We first predict part masks, followed by the affinity based grouping to hierarchically merge them into complete objects. We then pass these detections through the refinement layer to obtain the final predictions. We rank the predictions using the geometric mean of their predicted classification score $c$, box IoU $b$ and mask IoU $m$ from the refinement head as $s = \sqrt[3]{c * b * m}$.

| Cross-category setting | | | |
|---|---|---|---|
| Train On | Test On | # Seen classes | # Unseen classes |
| VOC | Non-VOC | 20 | 60 |
| LVIS | COCO | 1123 | 80 |

| Cross-dataset setting | | | |
|---|---|---|---|
| Train On | Test On | # Seen classes | # Unseen classes |
| COCO | UVO | 80 | open |
|  | ADE20k |  | 70 |
|  | OpenImagesV6 |  | 270 |

Tab. 1: **Evaluation settings**. Seen and unseen categories used in our evaluation.

| $VOC{\rightarrow}NonVOC$ | $AR_B^{100}$ | $AR_B^{300}$ | $AR_M^{100}$ | $AR_M^{300}$ |
|---|---|---|---|---|
| ***Bottom-up***(*No Training*) | | | | |
| SS | 14.3 | 24.7 | 6.7 | 12.9 |
| MCG | 23.6 | 30.8 | 19.4 | 25.2 |
| ***Top-down***(*Class-agnostic Training*) | | | | |
| MaskRCNN | 25.1 | 30.8 | 20.5 | 25.1 |
| Mask R-CNN$_{SS}$ | 24.1 | 24.9 | 20.9 | 21.7 |
| Mask R-CNN$_{SC}$ | 25.6 | 33.1 | 24.9 | 28 |
| ***Open-World Methods*** | | | | |
| OLN | 32.5 | 37.4 | 26.9 | 30.4 |
| LDET | 30.9 | 38.0 | 26.7 | 32.5 |
| GGN | 31.6 | 39.5 | 28.7 | 35.5 |
| UDOS | **33.5** | **41.6** | **31.6** | **35.6** |

| $LVIS{\rightarrow}COCO$ | $AR_B^{100}$ | $AR_B^{300}$ | $AR_M^{100}$ | $AR_M^{300}$ |
|---|---|---|---|---|
| MaskRCNN | 23.8 | 29.4 | 18.5 | 22.0 |
| Mask R-CNN$_{SC}$ | 21.3 | 27.9 | 17.9 | 24.2 |
| OLN | 28.5 | 38.1 | 23.4 | 27.9 |
| UDOS | **33.2** | **42.2** | **26.3** | **32.2** |

Tab. 3: **Cross-category generalization evaluation with large taxonomy**. All models are trained on 1123 categories from LVIS (excluding COCO categories), and evaluated on COCO 80 categories. UDOS outperforms OLN by 4.7% and 2.9% on box and mask $AR^{100}$.

Tab. 2: **Cross-category generalization evaluation on COCO.** Train on 20 VOC categories and test on 60 NonVOC categories. UDOS outperforms many competitive baselines as well as the current SOTA GGN on the VOC→NonVOC setting.

## 4 EXPERIMENTS

**Datasets and evaluations.** We demonstrate the effectiveness of UDOS for open-world instance segmentation under *cross-category* generalization within the same dataset, as well as *cross-dataset* generalization across datasets with different taxonomies (Tab. 1). We use the MS-COCO (Lin et al., 2014) for cross-category generalization and train the model using 20 categories from VOC and test on the 60 remaining unseen nonVOC classes following prior work (Kim et al., 2022; Saito et al., 2022; Wang et al., 2022). For cross-dataset generalization, we train on complete COCO dataset and directly test on validation splits of UVO (Wang et al., 2021b), ADE20k (Zhou et al., 2019) and OpenImagesV6 (Benenson et al., 2019) datasets without any fine-tuning. We also test large-taxonomy scenario by training on a subset of 1123 categories from LVIS (Gupta et al., 2019) and test on COCO. Both UVO and ADE20k datasets provide exhaustive annotations in every frame, which is ideal to evaluate open world models, while OpenImagesV6 with 350 categories allows to test our open world segmentation approach on large scale datasets.

**Implementation details.** We use Mask R-CNN model (He et al., 2017) with a ResNet-50-FPN (Lin et al., 2017) as our backbone. We train UDOS using SGD for 10 epochs with an initial learning rate of 0.02 on 8 GPUs. We use selective search (Uijlings et al., 2013) to generate unsupervised masks for images in COCO dataset. Note that the mask proposals are required *only during training*, and *not during inference* (Fig. 2). We follow prior works in open-world instance segmentation (Kim et al., 2022; Saito et al., 2022; Wang et al., 2022) and use average recall (AR) (between IoU thresholds of 0.5 to 1.0) as the evaluation metric. Since open world models generally detect many more objects in a scene than closed world models (see Fig. 5) and many datasets do not have exhaustive annotation, we use $AR^{100}$ and $AR^{300}$ as the evaluation metrics on both box ($AR_B$) and mask ($AR_M$) to avoid penalizing predictions of valid, yet unannotated, objects.

### 4.1 BASELINES

(i) **Image-computable masks**: We use masks generated by **MCG** (Pont-Tuset et al., 2016) and Selective Search (Uijlings et al., 2013) (**SS**), which are class-agnostic, learning-free proposal generation methods relying on low-level cues. (ii) **Mask-RCNN** (He et al., 2017) denotes Mask R-CNN training in class-agnostic fashion only on the seen classes, (iii) **Mask R-CNN**$_{SS}$ indicates Mask R-CNN trained using selective search proposals as the supervision instead of the ground truth anno-

| | COCO→UVO | | | | COCO→ADE20K | | | | COCO→OpenImages | | | |
|---|---|---|---|---|---|---|---|---|---|---|---|---|
| | $AR_B^{100}$ | $AR_B^{300}$ | $AR_M^{100}$ | $AR_M^{300}$ | $AR_B^{100}$ | $AR_B^{300}$ | $AR_M^{100}$ | $AR_M^{300}$ | $AR_B^{100}$ | $AR_B^{300}$ | $AR_M^{100}$ | $AR_M^{300}$ |
| MaskRCNN | 47.7 | 50.7 | 41.1 | 43.6 | 18.6 | 24.2 | 15.5 | 20.0 | 57.1 | 59.1 | 55.6 | 57.7 |
| Mask R-CNN$_{SS}$ | 26.8 | 31.5 | 25.1 | 31.1 | 18.2 | 25.0 | 17 | 21.6 | 34.0 | 42.7 | 33.1 | 38.8 |
| Mask R-CNN$_{SC}$ | 42.0 | 50.8 | 40.7 | 44.1 | 19.1 | 25.6 | 18.0 | 22.0 | 54.1 | 59.1 | 54.2 | 57.4 |
| OLN | 50.3 | 57.1 | 41.4 | 44.7 | 24.7 | 32.1 | 20.4 | 27.2 | 60.1 | 64.1 | 60.0 | 63.5 |
| LDET | 52.8 | 58.7 | 43.1 | 47.2 | 22.9 | 29.8 | 19.0 | 24.1 | 59.6 | 63.0 | 58.4 | 61.4 |
| GGN | 52.8 | 58.7 | 43.4 | 47.5 | 25.3 | 32.7 | 21.0 | 26.8 | 64.5 | 67.9 | 61.4 | 64.3 |
| UDOS | **53.6** | **61.0** | **43.8** | **49.2** | **27.2** | **36.0** | **23.0** | **30.2** | **71.6** | **74.6** | **66.2** | **68.7** |

Tab. 4: **Cross-dataset generalization evaluation for open world instance segmentation**. All models are trained on 80 COCO categories and evaluated on UVO (left), ADE20K (middle), Open-Images (right) as is without any fine-tuning.

tations, and (iv) **Mask R-CNN**$_{SC}$ denotes Mask R-CNN trained with BoxIoU and MaskIoU scoring to rank the proposals instead of the classification score.

We also compare with state of the art open-world instance segmentation algorithms OLN (Kim et al., 2022), LDET (Saito et al., 2022) and GGN (Wang et al., 2022). For fair comparison with UDOS, we use the result from GGN (Wang et al., 2022) *without* the OLN backbone.

### 4.2 UDOS OUTPERFORMS BASELINES ON CROSS-CATEGORY GENERALIZATION

Existing methods relying on bottom-up grouping with techniques like SS or MCG, or top-down architectures like Mask R-CNN trained on annotations for seen classes, struggle to effectively detect and segment instances from unseen classes. In contrast, UDOS overcomes these limitations (Tab. 2) by harnessing both ground truth annotations for known classes and unsupervised bottom-up masks for unknown classes, resulting in a significant improvement over all baseline methods, underscoring the effectiveness of our approach. From Tab. 2, UDOS achieves Box Average Recall (AR) of 33.5% and Mask AR of 31.6% setting a new state-of-the-art in cross-category open-world instance segmentation, outperforming the current state-of-the-art methods like GGN in both box and mask AR. Qualitative results in Fig. 5 and supplementary materials further illustrate how UDOS excels at detecting objects, even in cases where their annotations were missing from the training data.

Expanding training datasets to encompass larger taxonomies is a potential strategy for addressing the challenge of novel categories at test-time. However, our experiments, detailed in Tab. 3, reveal that this approach still falls short of achieving robust generalization to unseen categories. We leveraged the LVIS dataset (Gupta et al., 2019), which includes annotations for 1203 categories. During training, we excluded annotations with an IoU overlap greater than 0.5 with COCO masks, resulting in 79.5k instance masks from LVIS. When evaluated on COCO validation images, UDOS achieved 33.2% $AR_B^{100}$ and 26.3% $AR_M^{100}$, markedly outperforming the baseline methods. This underscores the effectiveness of UDOS in even handling datasets with large category vocabularies.

### 4.3 UDOS SETS NEW SOTA ON CROSS-DATASET GENERALIZATION

To provide a more realistic assessment of our model's open-world capabilities, we evaluated its performance on real-world target datasets like UVO, ADE20k, and OpenImages. These datasets contain a wide range of objects, including those not covered by COCO categories. It's worth noting that we refrained from fine-tuning our model on the target datasets or using any unlabeled target data during training. Results are in Tab. 4, and summarized below.

**COCO to UVO**. Since UDOS is designed to handle novel classes, it achieves much better performance than other baselines on the challenging UVO dataset that contains exhaustive annotations for every objects. UDOS clearly outperforms baseline approaches like Mask R-CNN (+5% $AR_B^{100}$). We also outperform OLN, LDET and GGN, setting the new state-of-the-art on the UVO benchmark.

**COCO to ADE20K** ADE20k (Zhou et al., 2019) is a scene parsing benchmark consisting of annotations for both *stuff* (road, sky, floor etc.) and discrete *thing* classes. We regard each annotation mask as a separate semantic entity and compute the average recall (AR) on both in-taxonomy and out-of-taxonomy objects to evaluate the ability of trained models to detect thing classes and group stuff classes in images. From Tab. 4, we observe that UDOS achieves box AR100 of 27.2% and mask AR100 of 23.0%, higher than all the baselines and other competing methods.

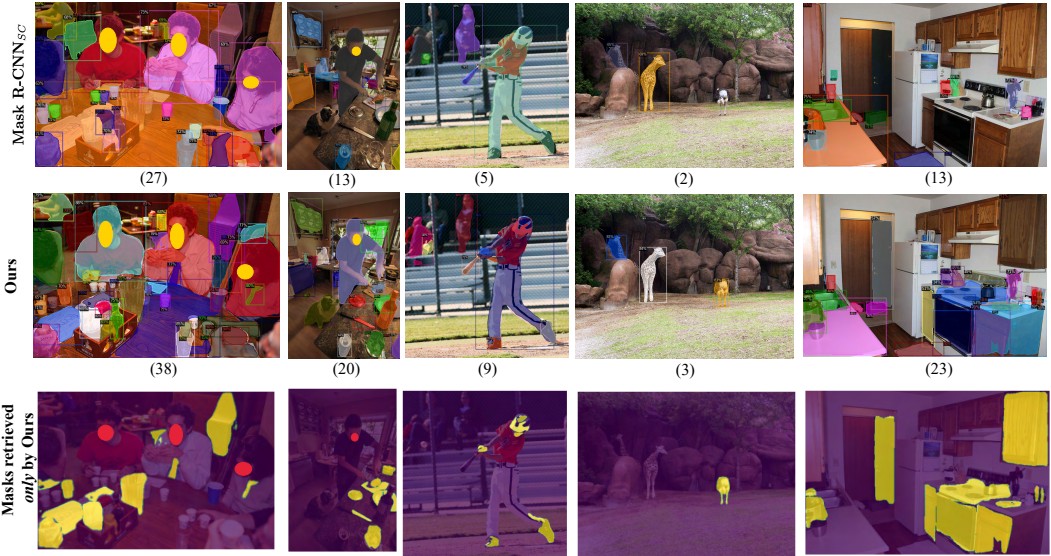

Fig. 5: **Visualization of segmentations for model trained only on VOC classes** from COCO dataset. The top row shows result using using Mask-RCNN$_{SC}$, second row shows output using UDOS and the third row shows some predictions made only by UDOS and missed by Mask-RCNN$_{SC}$. We also show the number of detections made by the network below each image. Starting from left most image, many classes like {jug, tissue papers, tie, eyeglasses}, {knife, cutting board, vegetables, glass}, {shoes, helmet, gloves}, {ostrich} and {dishwasher, faucet} among others which are not part of VOC-classes are missed by standard Mask-RCNN training, but detected using UDOS. More visualizations are provided in the supplementary.

(a)

| Group | Refine | $AR_B^{100}$ | $AR_M^{100}$ |
|---|---|---|---|
| ✗ | ✗ | 25.4 | 11.8 |
| ✓ | ✗ | 32.6 | 30.7 |
| ✓ | ✓ | **33.5** | **31.6** |

(b)

| BoxIoU | MaskIoU | $AR_B^{100}$ | $AR_M^{100}$ |
|---|---|---|---|
| ✗ | ✗ | 29.0 | 24.3 |
| ✓ | ✗ | 32.7 | 28.9 |
| ✗ | ✓ | 32.9 | 29.2 |
| ✓ | ✓ | **33.5** | **31.6** |

(c)

| Segmentation | $AR_B^{100}$ | $AR_M^{100}$ |
|---|---|---|
| Uniform Grid | 9.9 | 9.2 |
| SSN | 19.4 | 18.7 |
| Sel. Search | **33.5** | **31.6** |
| MCG | 32.4 | 29.4 |

(d)

Tab. 5: **Ablation results**. Effect of (a) grouping and refinement modules, (b), boxIoU and maskIou losses during training, (c) segmentation algorithm and (d) context dilation parameter $\delta$ on the VOC→NonVOC performance.

**COCO to OpenImagesV6** Again, UDOS consistently outperform all baselines as well as open-world methods like OLN and GGN by significant margins on the OpenImagesV6 dataset (Benenson et al., 2019). We achieve $AR_B^{100}$ of 71.6%, which is better than the strongest baseline Mask R-CNN by 14.5% and current state-of-the-art GGN by 7.1%. Likewise, $AR_M^{100}$ of 66.2% obtained by UDOS is 4.8% higher than GGN, setting new state of the art. Several visualizations of UDOS predictions on UVO, ADE20k and OpenImages datasets have been provided in the supplementary material.

## 4.4 ABLATIONS

We use the VOC to NonVOC cross-category generalization on COCO dataset for the ablations.

**Refinement and grouping modules** We show in Tab. 5a that without the proposed grouping and refinement modules, maskAR drops to 11.8% from 31.6%, as the masks are noisy and only correspond to parts of instances. Using a refinement module after grouping leads to more refined masks further improving the performance.

**Choice of proposal ranking** We show the importance of using BoxIoU and MaskIoU scoring functions in Tab. 5b, where significant drops in AR100 are observed without the use of both the scoring functions, validating the observations in prior works (Kim et al., 2022) that scoring module prevents over-fitting and improves open world learning.

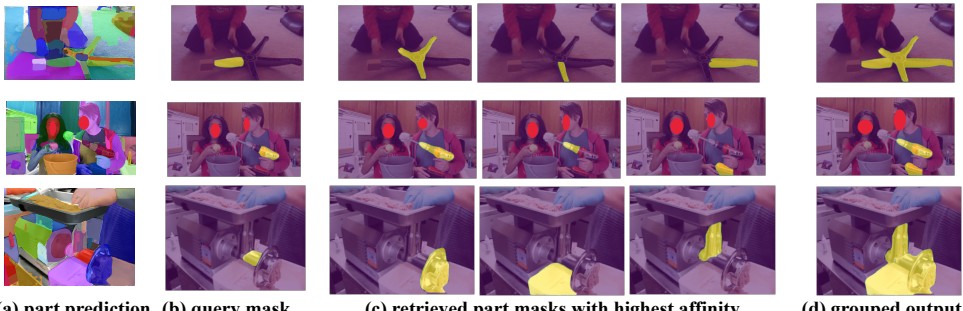

**(a) part prediction**   **(b) query mask**   **(c) retrieved part masks with highest affinity**   **(d) grouped output**

Fig. 6: **Visualization of pairwise affinity maps and grouped predictions.** Given a part mask as a query, we show the 3 nearest part masks of the query using our pairwise affinity. The images are taken from UVO dataset, and the affinity is computed using UDOS model trained on COCO. Our affinity-based grouping module correctly groups parts into whole instances even with unseen objects. The last row visualizes a failure case where the model retrieves a part mask from a neighboring instance.

**Influence of $\delta$** Intuitively, a small value of delta (part-mask expansion factor, equation 1) would not capture sufficient context around the region for extracting similarity while a very high value of $\delta$ would induce noisy features from different neighboring objects. In Tab. 5d, we show that a value of 0.1 achieves an optimum trade-off, so we use the same value of $\delta = 0.1$ in all our experiments.

**Choice of proposal generation** From Tab. 5c, we show that a naive segmentation of image using uniform grids by extracting $64{\times}64$ patches from the image expectedly performs worse, as these part masks do not semantically correspond to object parts. We also use super-pixels generated from SSN (Jampani et al., 2018), but found that bottom-up supervision generated from image-based segmentation algorithms like SS or MCG lead to much better accuracies.

**Visualizations of affinity maps** In Fig. 6, we present 3-nearest part masks retrieved for a given query mask using their affinity (equation 2) and the grouped outputs. We observe that different part masks of the same entity are often retrieved with high affinity, using our grouping module.

**Inference time comparison** Our grouping module is lightweight and adds negligible run-time overhead. Specifically, at 100 output proposals, MaskRCNN (He et al., 2017) and GGN (Wang et al., 2022) take 0.09s/im, MaskRCNN$_{SC}$ and OLN (Kim et al., 2022) take 0.12s/im while UDOS takes 0.13s/im (+0.01s/im) with stronger performance. Generating part-masks using selective search for the complete COCO (Lin et al., 2014) dataset takes around 1.5 days on a 16-core CPU, but we reiterate that the part-masks only need to be generated once before training and are not needed during testing/deployment (Fig. 2). We will publicly release the part-masks on COCO dataset along with the code and trained models.

## 5 DISCUSSION

In this paper, we conduct an investigation to understand what types of top-down learning generalize well in the context of open-world segmentation. Our observation revealed that learning from known classes to group part-level segmentations and learning to refine coarse boundaries are effective for generalizing to new categories. This allowed us to leverage classical bottom-up segmentation algorithms that provide class-agnostic yet coarse and over-segmented part masks within a top-down learning framework. We introduced UDOS, a novel approach that integrates top-down and bottom-up strategies into a unified framework. Our grouping and refinement modules efficiently convert part mask predictions into complete instance masks for both familiar and unfamiliar objects, setting UDOS apart from previous closed-world and open-world segmentation methods. Extensive experiments demonstrated the significant performance improvements achieved by UDOS across five challenging datasets, including COCO, LVIS, ADE20k, UVO, and OpenImages. We believe that UDOS can serve as a versatile module for enhancing the performance of downstream tasks, such as robotics and embodied AI, in handling unfamiliar objects in open-world scenarios.

**Limitations**. UDOS faces challenges in scenarios with densely clustered objects of similar appearance. A more robust, learnable grouping method, possibly trained with hard negatives, could enhance performance in such complex situations, suggesting a direction for future research.

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
