# OPEN-WORLD INSTANCE SEGMENTATION: TOP-DOWN LEARNING WITH BOTTOM-UP SUPERVISION
### SUPPLEMENTARY MATERIAL

Tab. 1: **Effect of weight sharing between RoI head and refinement head.** Comparison of results with and without sharing parameter weights between RoI module of part mask prediction head and the refinement head. Using separate heads for the RoI head of part-mask prediction module and refinement module improves $AR_M^{100}$ by 3.3%.

| | $AR_B^{100}$ | $AR_B^{300}$ | $AR_B^s$ | $AR_B^m$ | $AR_B^l$ | $AR_M^{100}$ | $AR_M^{300}$ | $AR_M^s$ | $AR_M^m$ | $AR_M^l$ |
|---|---|---|---|---|---|---|---|---|---|---|
| Shared weights | 33.5 | 41.3 | 13.2 | 43.6 | 60.6 | 28.3 | 33.6 | 11.4 | 37.7 | 48.4 |
| Non shared weights | 33.5 | 41.6 | 16.2 | 43.9 | 53.5 | 31.6 | 35.6 | 15.2 | 42.7 | 48.3 |

## 1 WEIGHT SHARING ON REFINEMENT MODULE

In UDOS, our design of the refinement module follows the RoI-heads of Mask R-CNN architecture. Specifically, our box and mask prediction heads in the refinement head have the same architecture as the box and mask prediction heads of the Mask R-CNN. Therefore, one option is to share the weights between the RoIHeads in the first stage Mask R-CNN and the refinement heads during training. However, we note from Table 1 that such an approach of weight-sharing between the two stages of UDOS hurts the mask accuracy on the cross-category detection on COCO. This is because the goals of prediction in the two stages are different. While the RoIHeads in the first stage are trained to predict part-masks and trained on weak supervision from bottom-up segmentation algorithms, the refinement head is trained only using ground truth annotations and is used to predict the final object boxes and masks. However, this improvement also comes with additional increase in model parameters from 57.4M to 86.5M. Also, we observed that using individual weights only benefits cross-category setting, while cross-dataset benefits from shared weights between the part-mask MaskRCNN and refinement head.

## 2 VISUALIZING OUTPUTS AT EACH STAGE OF UDOS

We visualize the outputs after each stage of UDOS for cross-category VOC to NonVOC setting in Fig. 1 and for cross-dataset setting in Fig. 2. We illustrate the effect of our part-mask prediction module in in generating the segmentation masks for parts of objects, rather than the whole objects. This enables us to detect a much larger taxonomy of objects than what are present in the annotated concepts. For example, in Fig. 1 for the case of cross-category transfer setting from VOC → Non-VOC, *tie* and *shoe* are not one of the annotated classes. Yet, our model effectively retrieves these from the image, instead of considering it a background or combining it with the boy. Our grouping module, powered by the context aggregation, then effectively groups the various part masks predicted on the *pot*, *boy* and *tie*. Note that the accuracy of predictions obtained by directly merging the part masks might be limited due to noisy part mask supervision, which are further corrected by our refinement layer. Similar observations for the cross-dataset setting are presented in Fig. 2.

## 3 QUALITATIVE COMPARISONS

In addition to the comparisons provided in the main paper, we provide more comparisons of predictions made by UDOS and Mask R-CNN$_{SC}$ in Fig. 3 for the setting where we train only using VOC categories from COCO. We also show the predictions made on the cross-dataset setting, by using a model trained on all COCO categories and testing on images from UVO Wang et al. (2021) in Fig. 4.

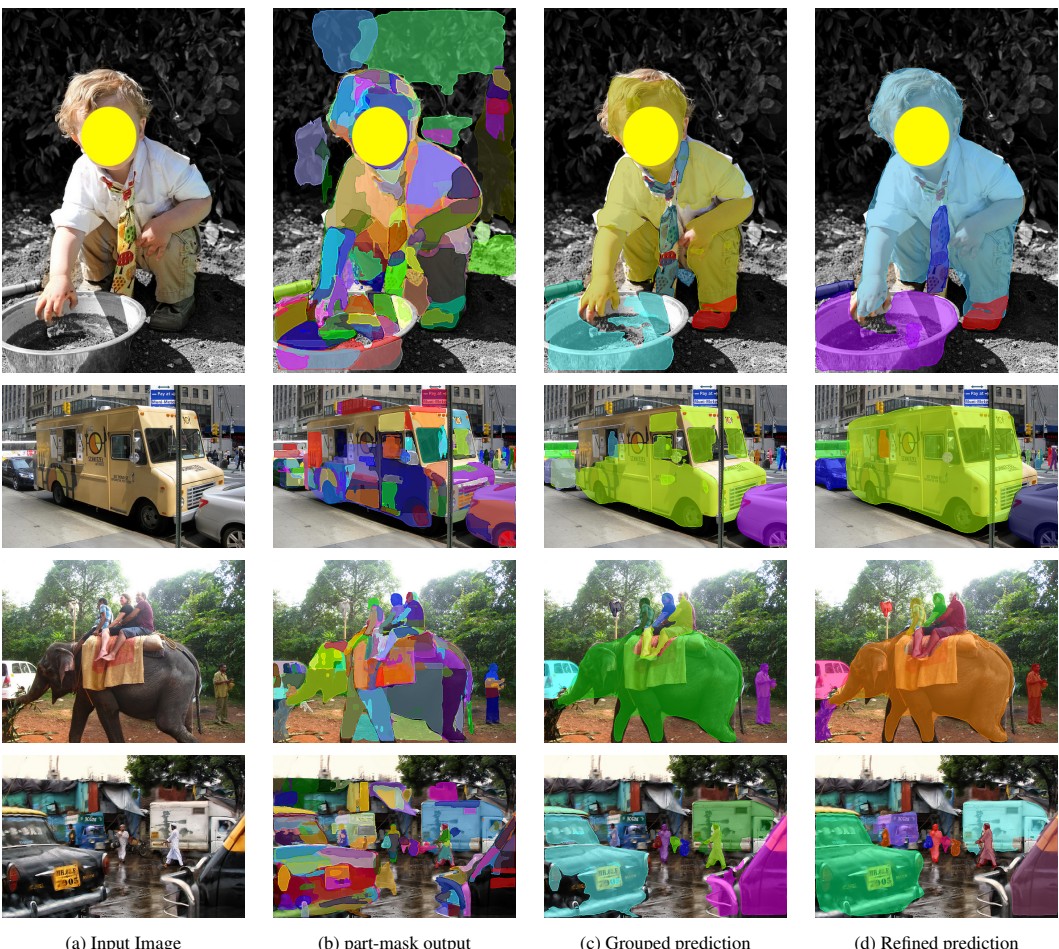

|  |  |  |  |
|---|---|---|---|
| (a) Input Image | (b) part-mask output | (c) Grouped prediction | (d) Refined prediction |

Fig. 1: **Visualizing outputs after each stage of UDOS for cross-category training**. All images belong to the COCO dataset, and outputs are generated using a model trained only on VOC categories. (a) shows the input image, followed by (b) Part-mask prediction, (c) grouped outputs using our affinity based grouping and (d) refined prediction. The masks in last two columns correspond to true-positives with respect to the ground truth. Note that classes such as *pot, van, elephant*, and *auto-rickshaw* do not belong to any of the training VOC categories. Also note that the merged outputs might be noisy due to the imperfection in the initial part-mask supervision used, which are corrected by our refinement layer.

In each case, we also show the predictions made *only* by UDOS and missed by Mask R-CNN$_{SC}$ (highlighted in yellow), indicating the utility of our approach on open world instance segmentation.

For instance, in the second column in Fig. 3, the predictions made by Mask R-CNN$_{SC}$ do not include objects like *keyboard*, *joystick*, *glass* and *speaker* which are efficiently retrieved by UDOS. Also note that the number of predictions made by UDOS is always higher than Mask R-CNN$_{SC}$ for both cases of cross-category transfer in Fig. 3 as well as cross-dataset transfer in Fig. 4.

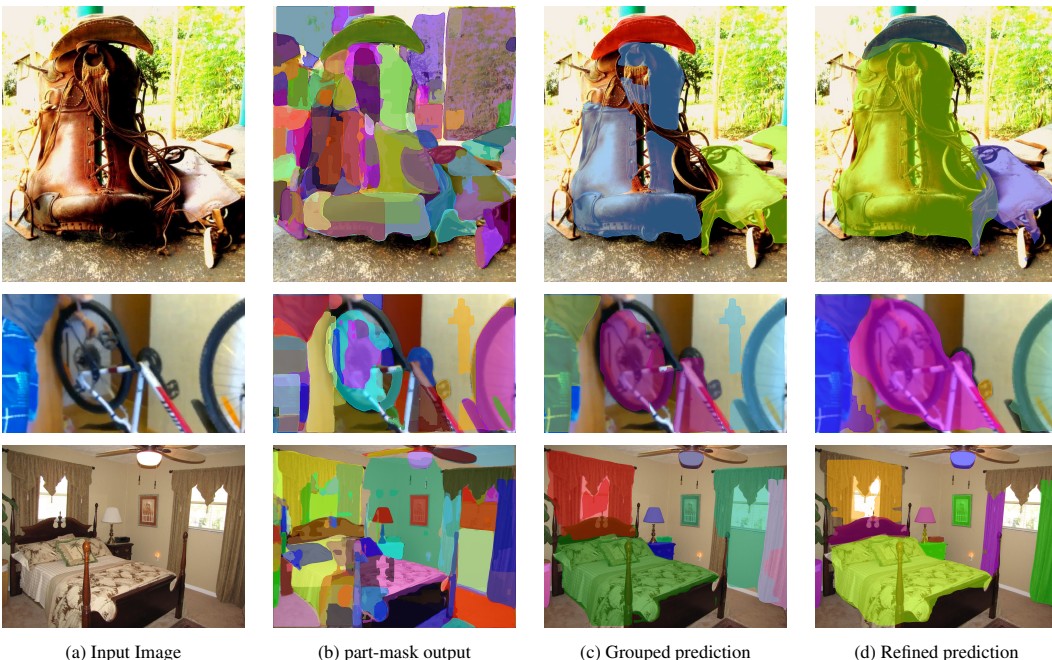

(a) Input Image      (b) part-mask output      (c) Grouped prediction      (d) Refined prediction

Fig. 2: **Visualizing outputs after each stage of UDOS for cross-dataset training**. The images in the three rows belong to OpenImages Benenson et al. (2019), UVO Wang et al. (2021) and ADE20K Zhou et al. (2019) datasets respectively. All outputs are generated by model trained on complete COCO dataset. (a) shows the input image, followed by (b) Part-mask prediction, (c) grouped outputs using our affinity based grouping and (d) refined prediction. The masks in last two columns correspond to true-positives with respect to the ground truth.

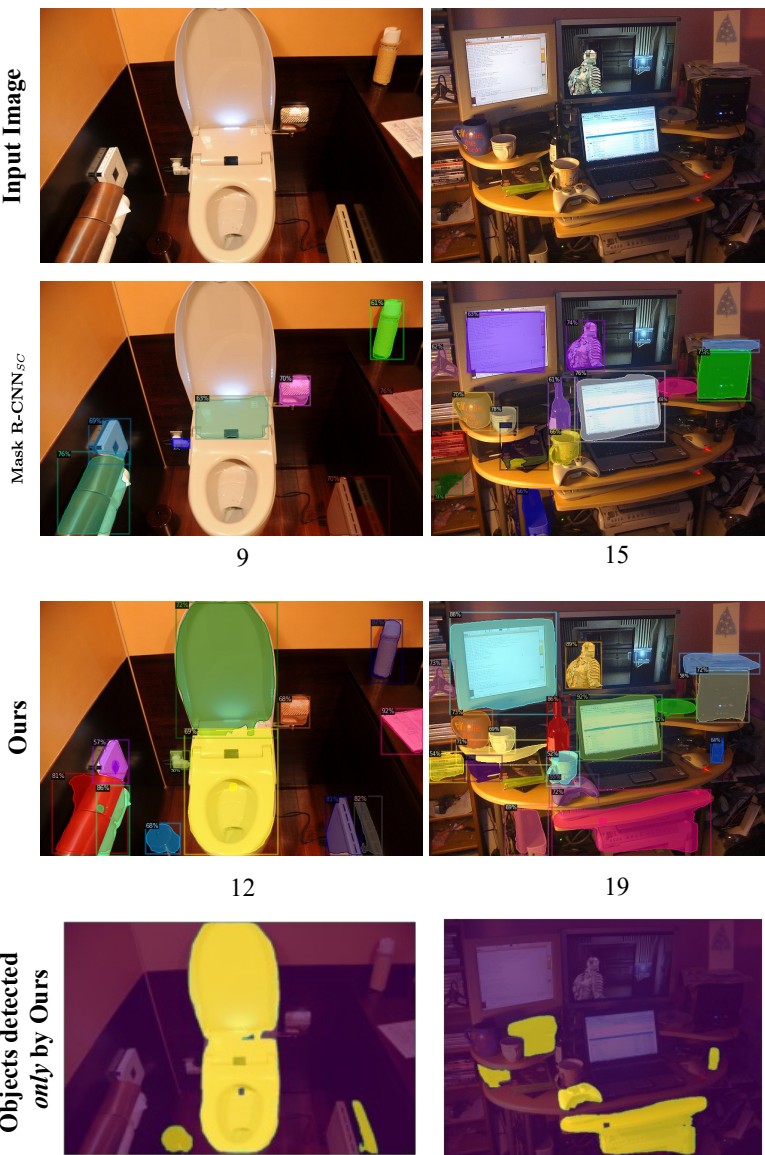

Fig. 3: **Visualization of segmentations for model trained only on VOC classes** from COCO dataset. For various input images given in the first row, the second row shows result using Mask-RCNN$_{SC}$, third row shows output using UDOS and the fourth row shows some predictions made only by UDOS and missed by Mask-RCNN$_{SC}$ on these images. We also show the number of detections made by the network below each image. All images belong to COCO dataset.

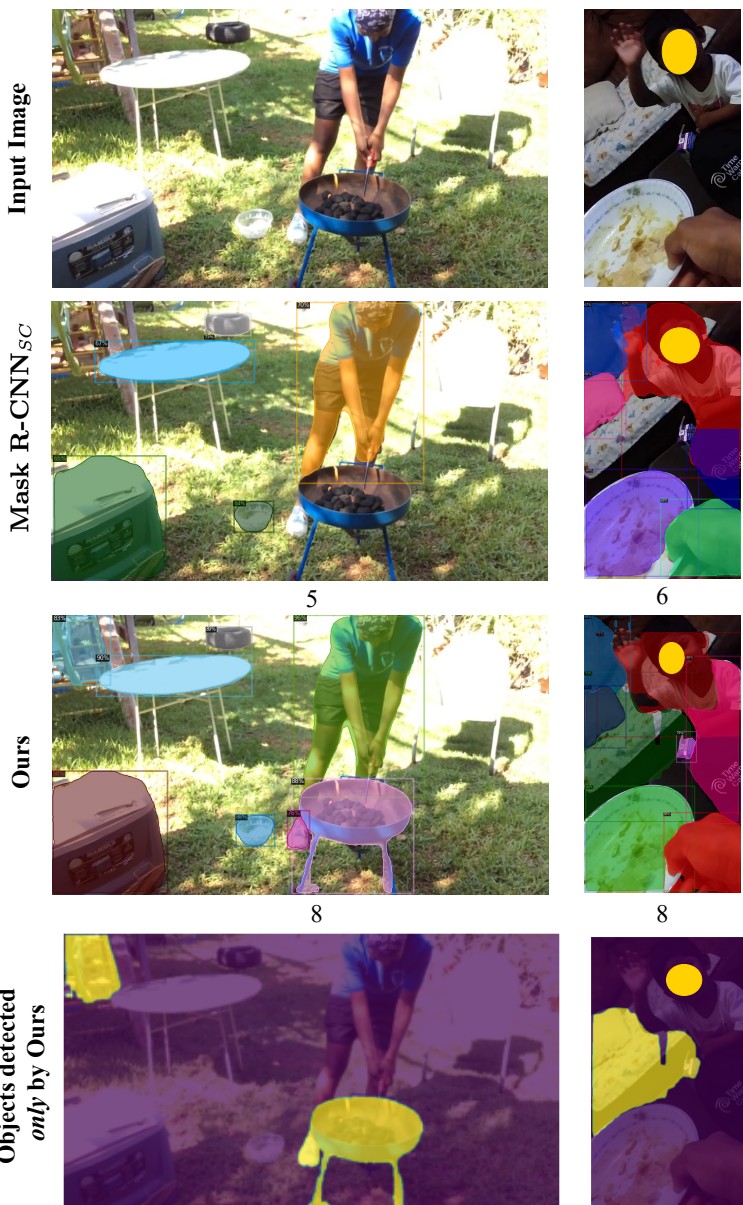

Fig. 4: **Visualization of segmentations for model trained on all COCO classes**. For various input images given in the first row, the second row shows result using Mask-RCNN$_{SC}$, third row shows output using UDOS and the fourth row shows some predictions made only by UDOS and missed by Mask-RCNN$_{SC}$ on these images. We also show the number of detections made by the network below each image. All images belong to UVO dataset.