# OpenReview forum: "Open-world Instance Segmentation: Top-down Learning with Bottom-up Supervision"
_ICLR.cc/2024/Conference — Submitted to ICLR 2024_

### Official Review · Reviewer_bvBC · 2023-10-23

**Soundness:** 3 good
**Presentation:** 3 good
**Contribution:** 2 fair
**Rating:** 5
**Confidence:** 5

**Summary:**

The paper addresses the limitations of top-down instance segmentation architectures in open-world scenarios, where predefined closed-world taxonomies may not be sufficient. To overcome this challenge, the authors propose a novel approach called Bottom-Up and Top-Down Open-world Segmentation (UDOS).

UDOS combines classical bottom-up segmentation methods within a top-down learning framework. It utilizes a top-down network trained with weak supervision derived from class-agnostic bottom-up segmentation to predict object parts. These part-masks are then refined through affinity-based grouping to generate precise instance-level segmentations.

The key advantage of UDOS is its ability to balance the efficiency of top-down architectures with the capacity to handle unseen categories by leveraging bottom-up supervision. By incorporating both approaches, UDOS achieves superior performance over state-of-the-art methods in cross-category and cross-dataset transfer tasks. The authors validate their approach on challenging datasets such as MS-COCO, LVIS, ADE20k, UVO, and OpenImages.

**Strengths:**

+ The paper demonstrates a high level of originality in several aspects. Firstly, it introduces the concept of combining classical bottom-up segmentation methods with a top-down learning framework to address the limitations of predefined taxonomies in open-world scenarios.

+ The use of weak supervision derived from class-agnostic bottom-up segmentation to predict object parts contributes to the originality of the proposed method.

**Weaknesses:**

- While the Multiscale Combinatorial Grouping (MCG) approach was proposed in 2016, it might be beneficial to consider the use of more recent methods, such as the Segmentation Attention Module (SAM), to enhance the generation of higher-quality masks for this problem. The integration of SAM into the existing framework could potentially improve the performance and accuracy of mask generation.

- In order to provide a comprehensive evaluation of the proposed approach, it would be valuable to compare it with relevant open-world panoptic segmentation techniques, such as ODISE (Open-vocabulary DIffusion-based panoptic SEgmentation). The inclusion of a comparative analysis with ODISE would enable a thorough assessment of the strengths and weaknesses of the proposed method and offer insights into its effectiveness in handling open-world scenarios.

**Questions:**

Please refer to paper Weakness.

---

> ### Author Response · Authors · 2023-11-23
> **Response to the reviewer**
>
> We thank the reviewer for their valuable feedback and highly relevant comments! We address each of them below.
>
> > **Using advanced super-pixel methods**
>
> Thank you for raising this important point. We use MCG as it is fast and efficiently scalable to generate initial masks for the datasets we used in our paper. Note that despite employing a basic superpixel method like MCG, our UDOS framework already outperforms the next best prior method by 2% maskAR100 for cross-category (Tab 2) and upto 5% maskAR100 in cross-dataset settings (Tab 4). We will update the paper with this discussion.
>
> > **Incorporating Segmentation Attention Module**
>
> We thank the reviewer for posing this idea. We unfortunately couldn’t find a canonical reference for the suggested “segmentation attention module”, closest works are [1] and [2] - both of which are trained on supervised, closed-world annotated images and hence not applicable to generate part-masks in an open-world setting.
>
> In case you meant Segment Anything Model (SAM)[3], we note that foundational models like SAM are reliant on millions of supervised images during training, making direct comparison of their open-world capabilities with ours difficult. Particularly, SAM's training data lacks category labels, making it infeasible to construct an evaluation set for categories not included in SAM's training data. However, UDOS is perfectly compatible to incorporate advances like SAM by replacing the initial super-pixel supervision with that generated by SAM, opening up exciting possibilities for enhanced open-world segmentation that builds upon UDOS in the future.
>
> [1] Jiang, Junzhe, et al. "DSA: Deformable Segmentation Attention for Multi-Scale Fisheye Image Segmentation." Electronics 12.19 (2023): 4059.
>
> [2] Gou, Yuchuan, et al. "Segattngan: Text to image generation with segmentation attention." arXiv preprint arXiv:2005.12444 (2020).
>
> [3] Kirillov, Alexander, et al. "Segment anything." ICCV  (2023).
>
>
> > **Comparison with ODISE**
>
> Thank you for bringing this prior work to our notice. We note that ODISE and UDOS are fundamentally different tasks. While ODISE is designed for open-vocabulary panoptic segmentation which requires labeling new classes in-the-wild, UDOS is focused on open-world instance segmentation which aims at detecting categories unseen during training (that is, objects with no masks during train time). Further, ODISE still employs seen and annotated instances for training the mask module using binary classification loss, which we already demonstrate does not generalize as well to open-classes as UDOS (MaskRCNN baseline in Tab 2 and Tab 4). We will include a discussion around this difference with ODISE in the updated version of the paper.

---

### Official Review · Reviewer_QToW · 2023-10-31

**Soundness:** 3 good
**Presentation:** 3 good
**Contribution:** 3 good
**Rating:** 6
**Confidence:** 4

**Summary:**

This paper proposes bottom-Up and top-Down Open-world Segmentation (UDOS), a novel approach that combines classical bottom-up segmentation methods within a top-down learning framework.

**Strengths:**

This method is reasonable and novel. Combining bottom-up and top-down is an interesting idea.

**Weaknesses:**

1. This paper generate candidate object regions through unsupervised segmentation methods. However, it cannot be guaranteed that these unsupervised methods can generate object regions that cover all regions. Especially when the number of categories increases, I question the performance of the unsupervised segmentation methods. The author should provide :1) the specific performance of the unsupervised segmentation methods, 2) experimental comparison with existing methods when categories are more, like COCO to LVIS.
2. The author should provide more result metrics with previous methods. For example, LDET also provides AP, AR10. The author should provide related performance comparisons to provide more comprehensive results.
3. [A] also proproses a CLN (region proposal generation algorithm). What's about performance comparision with this work.
4. What's about the details about Refinement module? I feel that this is all about previous methods, no matter the objectness ranking and inference.

[A] Detecting everything in the open world: Towards universal object detection. CVPR 2023

**Questions:**

Please refer to the weakness part. I will adjust the rating based on the author's feedback.

---

> ### Author Response · Authors · 2023-11-23
> **Response to the reviewer**
>
> We thank the reviewer for their highly insightful comments, for appreciating the novelty in UDOS and finding the work interesting! We answer the specific questions posed below.
>
> > **specific performance of the unsupervised segmentation methods**
>
> Thanks for raising this pertinent question! We show the segmentation performance of using only the unsupervised segmentation methods in Tab 2 of our paper. Specifically, the most competitive unsupervised method MCG yields 23.6 box AR100 compared to the most competitive MaskRCNN baseline of 25.6, indicating its effectiveness in detecting open-set instances.
>
> Also, we do agree that the unsupervised methods do not necessarily cover all the objects in the region, but as shown in our qualitative illustrations in Fig 5 in main paper and Fig 3,4 in the supplementary, UDOS does manage to detect many objects missed by MaskRCNN. This property is also reflected in the SOTA performances achieved by UDOS across the board (Tab 2 and Tab 4), with upto 5% improvements in maskAR100 compared to next best method.
>
> > **Effectiveness of UDOS on many-class settings**
>
> We note that the UVO dataset used in our paper is exhaustively annotated to cover every object in the image across several types of categories and scenes on a large-scale. It is notable that many of these categories are not even covered in the 1.2k sized LVIS taxonomy at all, as shown in [1]. On this dataset, UDOS achieves SOTA results outperforming all prior methods (COCO -> UVO, Table 4), effectively demonstrating UDOS's capabilities in handling open-world scenarios where a wide range of unseen objects may be present.
>
> > **more result metrics**
>
> Using AP/AR10/AR100, UDOS gives 2.8/15.0/31.6 while LDET gives 5.0/16.3/27.4 on the VOC to NonVOC setting. However, note that on COCO with non-exhaustive annotations, evaluating precision (AP) or low AR@K (like AR10), may unfairly punish detecting valid, yet un-annotated objects. We remark that other prior works in open-world segmentation (including OLN and GGN) also often choose not to report AP on cross-category generalization for this reason. We will add this point to the paper, along with the AP result.
>
> > **Comparison with CLN**
>
> We thank the reviewer for bringing this paper to our attention. While the region proposal generation algorithm in CLN employs label fusion from multiple closed-world datasets to mimic an open-world setting, this approach may not guarantee true open-world capabilities, as classes not included in any training datasets could still be overlooked. In contrast, our annotation-free and vocabulary-free segmentation method utilizes bottom-up low-level grouping to achieve state-of-the-art open-world capabilities with a remarkably simple framework. We will add this citation and relevant discussion to the paper.
>
> > **Details about the refinement module.**
>
> Our refinement method follows the mask and box prediction heads from previous methods only in design, but significantly differs in the purpose served. Specifically, prior methods feed the region proposals for mask prediction, while we feed the grouped part-masks as input to the refinement module for correcting noisy part-mask groupings. The significance of our refinement module is highlighted qualitatively in Tab.5a (+1% improvement in AR100) and qualitatively in Fig. 1 and Fig 2 in the supplementary through several visualizations.  It is also notable that just adding this refinement head to baseline MaskRCNN (_MaskRCNN_sc_ in Tab 2 and Tab 4) without the bottom-up grouping significantly underperforms UDOS by atleast 8% in AR value.
>
> [1] Wang, Weiyao, et al. "Unidentified video objects: A benchmark for dense, open-world segmentation." _Proceedings of the IEEE/CVF International Conference on Computer Vision_. 2021.

---

### Official Review · Reviewer_Rvq9 · 2023-11-01

**Soundness:** 2 fair
**Presentation:** 3 good
**Contribution:** 2 fair
**Rating:** 5
**Confidence:** 5

**Summary:**

The paper proposed the UDOS for open-world instance segmentation that combines bottom-up unsupervised grouping with top-down learning. This model designed a grouping module and refinement method to achieve SOTA performance on multiple datasets.

**Strengths:**

The group-parts-to-whole strategy for segmentation is interesting.
Experiments on multiples datasets verify the effectiveness of the proposed methods.
The paper writing and organization are good and clear.

**Weaknesses:**

Question:
1. Is there any time-consuming experiments on the cluster in the grouping module? Because the similarity is calculated two-by-two.
2. I am interested in the AP performance if adding the classification head in cross-datasets and cross-category setting. I know the task is category-free and different from open-vocabulary segmentation task, but I wander the segmentation performance with higher recall.
3. As we know, the segment anything (SAM[1]) has high generalizability in category-free segmentation task. It is a foundation model pretrained in many data, but its zero-shot ability is strong without fine-tune in specific datasets in category-free segmentation task, so I think the comparison is necessary. Can this have higher recall that SAM? If not, please discuss on the contribution.
4. Why exclude part masks from U that overlap with any ground truth mask in S with an IoU greater than 0.9? Please discuss on it with experiments.
5. How about the grouping result on these situations: two same-category instances are close (or overlap), two instance with similar color, two hierarchical categories (e.g. clothes and person).
[1] Segment Anything, ICCV2023.

**Questions:**

See weakness.

---

> ### Author Response · Authors · 2023-11-23
> **Response to the reviewer**
>
> We thank the reviewer for their very useful feedback and raising several pertinent questions! We address each of them below.
>
> > **Training and inference times for the grouping module.**
>
> Our UDOS framework utilizes a fast and efficient agglomerative clustering algorithm from scikit-learn, *ensuring minimal time and memory overhead* even when handling hundreds of part-masks. While standard MaskRCNN training on 8 Nvidia A10 GPUs takes about 22 hours, UDOS with its grouping and refinement module takes less than 24 hours. As already noted in the paper (sec 4.4), UDOS takes 0.13sec/image (7 FPS) during inference compared to MaskRCNN's 0.09 sec/image (11 FPS), while delivering remarkably higher accuracy.
>
> > **AP Values**
>
> UDOS gives an AP value of 2.8 while LDET gives 5.0 on the VOC to COCO setting on the binary mask classification. However, evaluating precision using AP on COCO with non-exhaustive annotations may unfairly punish detecting valid, yet un-annotated objects. We remark that other prior works in open-world segmentation (including OLN and GGN) also often choose not to report AP on cross-category generalization for this reason. We will add this point to the paper, along with the AP result.
>
> > **Comparison with Segment Anything (SAM)**
>
> This is a great question, thanks for raising this comment! Although foundational models like SAM expand the potential for segmentation, their reliance on vast amounts of supervised images during training impedes a fair comparison of their open-world capabilities with ours. Particularly, SAM's training data lacks category labels, making it infeasible to construct an evaluation set for categories not included in SAM's training data. However, note that UDOS is perfectly compatible to incorporate advances like SAM by replacing the initial super-pixel supervision with that generated by SAM, opening up exciting possibilities for enhanced open-world segmentation. We will add a citation to SAM, along with this discussion, in the final version.
>
> > **Excluding masks in U that overlap with S.**
>
> We remove part masks from U with significant overlap with those in the annotated set S to avoid possibly redundant supervision. Removing these masks using a threshold of 0.9 made no change to the resulting AR values, hence is optional. We will add this remark to the paper.
>
> > **UDOS capabilities in various situations.**
>
> This is a great point raised by the reviewer! We thank you for suggesting a novel protocol to construct evaluations, we believe that following these suggestions and conducting large-scale studies on failure modes of existing systems in the open-world can be a new contribution in itself! In current setting, despite the difficulty in quantitative evaluations, we still aim to provide understanding of these cases qualitatively in the main paper and the supplementary through several visualizations where UDOS shows its prominence in all those cases, such as:
>
> - _Similar category objects close to each other_: The glasses on the table (Fig 5 in main paper) and the people on the elephant (Fig 1 in the supplementary) are segmented with separate masks even when they are close together.
> - _Similar colored instances_: The two cabs (Fig 1 in the supplementary) are segmented separately.
> - _Hierarchical categorization_: The dress and the boy (Fig 1 in the supplementary), the uniform and the baseball player (Fig 5 in main paper) as well as the truck and its wheel (Fig 1 in the supplementary) are segmented together. Note that it is in general difficult to resolve this ambiguity in an open-world setting without the knowledge of real world categorization of the class hierarchies.

---

### Official Review · Reviewer_jfRf · 2023-11-02

**Soundness:** 1 poor
**Presentation:** 3 good
**Contribution:** 1 poor
**Rating:** 3
**Confidence:** 4

**Summary:**

This paper proposes a top-down bottom-up approach for open-set instance segmentation. The bottom-up segmentation module is used to predict object parts, and then the authors use a clustering/group method to assemble parts into objects.  The main point that the authors try to argue is that, this bottom-up module somehow fits well in the open-world (open-set) instance segmentation scenario.

**Strengths:**

originality: The approach involves quite a few components. To me the authors build a quite complex system and it's unclear to me what the motivation and which component is the main reason which contributes to the good performance.

quality: borderline

clarity: The idea is clear and the paper is easy to follow

significance: The task per se is quite important. However I do not think the system presented in this paper is good enough to have an impact on open-world instance segmentation.

**Weaknesses:**

1) The bottom-up module is quite complex, involving a few components. I do see the authors did ablation experiments to justify some design choices, it is not clear why  part-segmentation and grouping work better than other baseline approaches.  Part-segmentation + grouping appeared in the literature long time ago and researchers abandoned this idea.  Current experiments in this paper do not convince me that this is actually a better idea for open-world segmentation.  A simple baseline will be to train a class-agnostic instance segmentation using, e.g. COCO annotations.  Papers already showed that a  class-agnostic model works better for open-world problems.

2) The compared methods are very old. For example, authors choose Mask RCNN and MCG as the baseline methods. These two methods are very old. The authors will need to consider recent methods. Even for top-down methods, Mask2former etc. will be a much better choice. I see that the authors might argue that the proposed method can use any other top-down method to replace Mask RCNN. But still why choose MaskRCNN in the first place. Using a more recent method will make the experiment results more convincing.

**Questions:**

See above

---

> ### Author Response · Authors · 2023-11-22
> **Response to the reviewer**
>
> We thank the reviewer for their valuable feedback on our paper! We address each of their questions below!
>
> > **why should part-segmentation and grouping be preferred compared to class-agnostic training?**
>
> Firstly, the suggested class-agnostic instance segmentation is *already included as one of our baselines* (named _MaskRCNN_, and related variants), for cross-category setting (using VOC-COCO annotations) in Table 2 and cross-dataset setting (using all COCO annotations) in Table 4. Despite being a strong baseline, the class-agnostic models still suffered from weaker performance in open-world instance segmentation literature as also shown by other works in this field in OLN, LDET and GGN. We emphasize that UDOS *outperforms this exact baseline suggested by the reviewer by 8.4%* in cross-category and *6% in cross-dataset* settings. We refer the reviewer to Table 2, Table 4 and section 4.1 where we explicitly describe the baselines, including the class-agnostic model, a few improved variants of it as well as the state-of-the-art methods from recent literature.
>
> > **Prior works in part-segmentation + grouping**
> 1. Reviewer mentions that part-segmentation and grouping methods appeared in literature a long time ago but failed to provide any specific references. Thus, we cannot provide a detailed comparison. We do, however, like to refer reviewers to our related work section on how our work is related to prior works for segmentation.
> 2. In addition, the reviewer mentions that the idea of part-segmentation + grouping has been attempted before and has been abandoned. We respectfully disagree that this is a weakness and a justification for the low rating of our work. Firstly, we emphasize that our approach stands out from prior works due to two novel design choices: (i) *employing weak supervision from bottom-up part segments* for handling unseen instances and (ii) *integrating a refinement module* within an end-to-end framework for improving noisy part-mask groupings. These contributions *are validated both quantitatively (Table 5a and 5c) and qualitatively (Figure 5 and supplementary Figures 3-4)* in our paper. Second, we do not believe that unsuccessful prior attempts with conceptually similar ideas diminishes the value of new works. In fact, our strong empirical performance suggests that combining part segmentation and grouping indeed achieves strong performance in the open-world.
>
> > **Current experiment setting is not convincing**
>
> We thank the reviewer for noting their concern. We note that the current experiments and baselines in the paper are very comprehensive, covering diverse datasets (COCO, LVIS, UVO, ADE and OpenImages) and transfer settings (cross-category and cross-dataset), which were also used in all prior works in open-world instance segmentation (LDET, OLN, GGN). *UDOS comprehensively outperforms all these methods on all the settings*, thus highlighting the applicability of our approach for open world segmentation. In addition, the ablations clearly justify several of our design choices, such as the use of part-mask supervision or the grouping and refinement modules.
>
> > **Choice of MaskRCNN as the backbone**
>
> We employ MaskRCNN to enable fair comparisons with prior works such as LDET, GGN and OLN - all of which use MaskRCNN for open-world segmentation. We also highlight that despite employing a basic top-down architecture like MaskRCNN and superpixel method like MCG, our *UDOS framework already outperforms the best prior method* by 2% maskAR100 for cross-category (Tab 2) and upto 5% maskAR100 in cross-dataset settings (Tab 4). Moreover, as rightly noted, UDOS framework is designed to integrate seamlessly with any segmentation architecture, enabling it to incorporate recent and future advancements to potentially improve results further.

---

### Meta-Review · Area_Chair_6ENu · 2023-12-22

**Metareview:**

The proposed bottom-Up and top-Down Open-world Segmentation (UDOS) method makes use of unsupervised/bottom-up grouping during training to learn part segmentation, grouping, and refinement modules alongside the standard apparatus for supervised instance segmentation. In doing so, the method seeks to learn a segmentor that generalizes better to open-world data because the modules for parts, groups, and refinement may be less class-specific given their training on bottom-up segments. These additional modules add little to the computational overhead of the method, as measured by timing the difference at ~1/100 s. Experiments cover a variety of cases in the cross-category and cross-dataset settings, although they are restricted to Mask R-CNN as a base method, and to COCO as the sole source dataset for transfer. Across the cases evaluated the proposed UDOS delivers an accuracy improvement boost of 2-4 points.

Expert reviewers are divided with borderline/positive (QToW), borderline/negative (Rvq9, bvBC), and reject (jfRf) ratings. All have expertise in detection, instance segmentation, and open-world recognition. The authors provide a rebuttal, but only a single reviewer responds, and only a single additional reviewer participates in the reviewer-AC discussion phase. Both reviewers in the discussion confirm their borderline ratings, though QToW explains they are not opposed to rejection. The AC has considered each review and rebuttal carefully, to compensate for the lack of discussion, and has read the submission itself closely.

The AC sides with rejection. The submission does not satisfy reviewers' requests for (1) a more recent basis for experiments, (2) experiments on certain transfer pairs like COCO to LVIS, and (3) generally a need for more analysis especially of failure modes (and success modes for that matter). However, certain comments on weaknesses were downweighted in this decision: the complaint about part segmentation + grouping as "old" was not grounded by citations in the context of open-set recognition (or any at all), and the calls for comparison with the concurrent work SAM at ICCV'23 (Rvq9), CLN at CVPR'23 (QToW), and ODISE at CVPR'23 (bvBC) are fine as suggestions but are not weaknesses under the ICLR 2024 policy on contemporaneous work. Although the submission does report benchmark progress, this is not a sufficient condition for publication, as the empirical results have not convinced the reviewers and provided informative insight to guide future projects.

Strengths

- There is general agreement that incorporating bottom-up segmentation into top-down instance segmentation is of interest and could potentially help for open-world recognition. The experimental results do show this for the chosen benchmarks and baselines.
- The state-of-the-art accuracies for the particular scope of cross-category and cross-dataset comparisons for the given datasets (COCO VOC-to-Non-VOC, COCO-to-UVO, COCO-to-ADE20k, and COCO-to-OpenImages6).
- The variety of ablations for the components, design choices, and hyperparameters (Sec. 4.4) give some measure of the sensitivity of the proposed method for the adopted basis of Mask R-CNN.

Weaknesses

- The proposed UDOS is an assemblage of prior methods without technical innovation, such as MCG, Mask RCNN, the scoring of Kim et al. '22, and the refinement module (QToW, jfRf).
- The results do not include all of the metrics of interest to reviewers that are present in existing work (Rvq9, QToW), specifically AP. This is provided in the rebuttal with better results for LDET than UDOS, and while there is reason to be uncertain of the metric given unannotated objects, there is not consensus across papers on this point.
- It is not clear if the results would hold with current detection and segmentation methods. Although one could argue the proposed UDOS is conceptually general, its concrete choices of base detection method (Mask R-CNN) and bottom-up segmentation methods (SS, MCG) are significantly behind the state-of-the-art at this point. To be clear, the fair comparison with existing methods that rely on Mask R-CNN is valuable. That said, these experiments not sufficient: results on more backbones and datasets would strengthen the submission to clear the bar.

**Justification For Why Not Higher Score:**

- The experiments do not meet the community where it is w.r.t. current methods and datasets. Multiple reviewers requested results for base methods beyond Mask R-CNN and while UVO is a recent benchmark it is not the only benchmark of interest as LVIS was likewise requested. There is a scientific need to report the results included in the submission, as existing methods are based on Mask-RCNN, but more comprehensive and convincing results would include a ViT based method for example.
- The lack of technical novelty limits the potential impact for the work, and it is not made up for by empirical novelty or comprehensiveness. Experimenting with and analyzing a method assembled from existing approaches is not grounds for rejection, but it must justify itself by informing and convincing its audience (and the reviewers, as a sample of the potential audience) with the results that it has. That said, reviewers did appreciate the motivation for bottom-up and top-down segmentation, so further innovation in the generation or merging of the bottom-up component could be incorporated into a future resubmission.

**Justification For Why Not Lower Score:**

N/A

---

### Decision · Program_Chairs · 2024-01-16

Reject